# The Calculation and Evaluation of an Ultrasound-Estimated Fat Fraction in Non-Alcoholic Fatty Liver Disease and Metabolic-Associated Fatty Liver Disease

**DOI:** 10.3390/diagnostics13213353

**Published:** 2023-10-31

**Authors:** Pál Novák Kaposi, Zita Zsombor, Aladár D. Rónaszéki, Bettina K. Budai, Barbara Csongrády, Róbert Stollmayer, Ildikó Kalina, Gabriella Győri, Viktor Bérczi, Klára Werling, Pál Maurovich-Horvat, Anikó Folhoffer, Krisztina Hagymási

**Affiliations:** 1Department of Radiology, Medical Imaging Center, Faculty of Medicine, Semmelweis University, Korányi S. u. 2., 1083 Budapest, Hungary; zsombor.zita@stud.semmelweis.hu (Z.Z.); ronaszeki.aladar.david@semmelweis.hu (A.D.R.); budai.bettina@med.semmelweis-univ.hu (B.K.B.); csongrady.barbara@stud.semmelweis.hu (B.C.); stollmayer.robert@stud.semmelweis.hu (R.S.); kalina.ildiko@semmelweis.hu (I.K.); gyori.gabriella@semmelweis.hu (G.G.); berczi.viktor@semmelweis.hu (V.B.); maurovich-horvat.pal@med.semmelweis-univ.hu (P.M.-H.); 2Department of Surgery, Transplantation and Gastroenterology, Faculty of Medicine, Semmelweis University, Üllői út 78., 1082 Budapest, Hungary; werling.klara@med.semmelweis-univ.hu (K.W.); hagymasi.krisztina@med.semmelweis-univ.hu (K.H.); 3Department of Internal Medicine and Oncology, Faculty of Medicine, Semmelweis University, Korányi S. u. 2/A., 1083 Budapest, Hungary; folhoffer.aniko@med.semmelweis-univ.hu

**Keywords:** attenuation coefficient, backscatter-distribution coefficient, liver stiffness, quantitative ultrasound, non-alcoholic fatty liver disease, metabolic-associated fatty liver disease, hepatic steatosis, ultrasound-estimated fat fraction, proton density fat fraction

## Abstract

We aimed to develop a non-linear regression model that could predict the fat fraction of the liver (UEFF), similar to magnetic resonance imaging proton density fat fraction (MRI-PDFF), based on quantitative ultrasound (QUS) parameters. We measured and retrospectively collected the ultrasound attenuation coefficient (AC), backscatter-distribution coefficient (BSC-D), and liver stiffness (LS) using shear wave elastography (SWE) in 90 patients with clinically suspected non-alcoholic fatty liver disease (NAFLD), and 51 patients with clinically suspected metabolic-associated fatty liver disease (MAFLD). The MRI-PDFF was also measured in all patients within a month of the ultrasound scan. In the linear regression analysis, only AC and BSC-D showed a significant association with MRI-PDFF. Therefore, we developed prediction models using non-linear least squares analysis to estimate MRI-PDFF based on the AC and BSC-D parameters. We fitted the models on the NAFLD dataset and evaluated their performance in three-fold cross-validation repeated five times. We decided to use the model based on both parameters to calculate UEFF. The correlation between UEFF and MRI-PDFF was strong in NAFLD and very strong in MAFLD. According to a receiver operating characteristics (ROC) analysis, UEFF could differentiate between <5% vs. ≥5% and <10% vs. ≥10% MRI-PDFF steatosis with excellent, 0.97 and 0.91 area under the curve (AUC), accuracy in the NAFLD and with AUCs of 0.99 and 0.96 in the MAFLD groups. In conclusion, UEFF calculated from QUS parameters is an accurate method to quantify liver fat fraction and to diagnose ≥5% and ≥10% steatosis in both NAFLD and MAFLD. Therefore, UEFF can be an ideal non-invasive screening tool for patients with NAFLD and MAFLD risk factors.

## 1. Introduction

Non-alcoholic fatty liver disease (NAFLD) is a highly prevalent and potentially serious condition affecting up to 30% of the population, and it is also the most common cause of chronic liver disease (CLD) worldwide [1]. NAFLD is characterized by the pathologic accumulation of fat in the liver with a proton density fat fraction, which can be detected with MRI (MRI-PDFF) in excess of 5.6% or the accumulation of lipid droplets in more than 5% of hepatocytes and detected with liver biopsy [2]. It can range from simple, benign fatty liver disease (FLD) to non-alcoholic steatohepatitis (NASH), which can lead to cirrhosis and liver failure. NAFLD is also a diagnosis of exclusion that can be applied only to patients without known chronic liver disease, or a recent history of alcohol abuse. Meanwhile, in addition to demonstrating hepatic steatosis, the diagnosis of metabolic-associated fatty liver disease (MAFLD) is based on positive criteria, such as the presence of obesity, metabolic dysregulation, and type 2 diabetes. The exclusion of other disease etiologies is not a prerequisite of the MAFLD diagnosis [3].

Accurate diagnosis and monitoring of fatty liver disease are crucial for early intervention and prevention of complications. While liver biopsy has traditionally been the gold standard for diagnosis, it is invasive, carries risks, and samples only a small portion of the liver [4]. Non-invasive techniques are increasing in popularity for the detection of hepatic steatosis. Although very precise and recommended as a reference method for fat quantification, MRI-PDFF is expensive, and its availability is also limited [5]. Quantitative ultrasound (QUS) has emerged as a promising non-invasive technique for the diagnosis of fatty liver disease [6,7]. Artificial intelligence (AI) aided the diagnosis of hepatic steatosis with an automated measurement of the hepatorenal brightness index (HRI), which has also become available [8].

Among the QUS parameters, the ultrasound attenuation coefficient (AC) and backscatter-distribution coefficient (BSC-D) have been extensively validated for the detection of hepatic steatosis in NAFLD [9,10,11]. The AC attenuation coefficient measures the rate at which ultrasound waves weaken as they propagate through liver tissue [12]. The BSC-D quantifies the amount of ultrasound energy that is scattered back toward the transducer after interacting with tissues [13]. Both methods have been able to detect grade 1 and grade 2 steatosis with good to excellent accuracy, mitigating the need for a liver biopsy or other more expensive imaging methods [10,14]. On the other hand, a significant drawback is that, due to technical variations, QUS parameters developed by different vendors may have different diagnostic thresholds, making a direct comparison between measurements at multiple sites difficult [15,16]. Also, both AC and BSC-D suffer from the exponential loss of signal in higher grades of steatosis, resulting in the saturation of the dynamic range and a non-linear relationship between QUS parameters and the liver fat fraction [8,17,18]. In a couple of recently published studies, multivariable regression models were tested for the prediction of MRI-PDFF from QUS parameters [17,18]. The conversion of QUS parameters to the fat fraction also improves comparability in a clinical setting.

In this study, we aimed to perform a systematic evaluation of the relationship between AC, BSC-D, liver stiffness (LS), and steatosis measured with MRI-PDFF. We have developed a new prediction model using non-linear least squares analysis to calculate the ultrasound-estimated fat fraction (UEFF) based on a combination of AC and BSC-D parameters. To the best of our knowledge, we are the first to demonstrate that UEFF can be used for diagnosing NAFLD-related and MAFLD-related hepatic steatosis with similar accuracy.

## 2. Materials and Methods

### 2.1. Patient Selection

This single-center retrospective study was approved by the Institutional and Regional Science and Research Ethics Committee of our university. The patients provided written informed consent for QUS and MRI scans. All procedures and data processing were performed in compliance with the World Medical Association Declaration of Helsinki, revised in Edinburgh in 2000. We retrospectively collected data on patients with suspected fatty liver disease who had QUS, shear wave elastography (SWE), and quantitative MRI scans between July 2020 and May 2023 from our institution’s picture archiving and communication system (PACS). The patients’ demographics, the results of laboratory tests, and medical history were collected from electronic medical reports (Table 1).

The inclusion criteria in the NAFLD group were 18 years or older, signed informed consent, valid QUS, SWE, and MRI-PDFF measurements, and clinical findings consistent with NAFLD according to the European Association of Studying the Liver (EASL) guidelines [2]. The MAFLD group contained patients with clinical findings consistent with MAFLD according to the international expert consensus, excluding the NAFLD group [3]. Patients in the NAFLD group did not report significant—over 20 g (2 drinks) for females or 30 g (3 drinks) for males—daily alcohol consumption in the previous two years and did not have a known history of chronic liver disease. Meanwhile, patients in the MAFLD group had at least one of the three conditions: obesity, type 2 diabetes, metabolic dysregulation, and known chronic liver disease. Three patients in the MAFLD group reported significant ongoing or recent alcohol consumption. Patients with acute liver failure, acute-on-chronic liver failure, and extrahepatic biliary obstruction were excluded from this study. We also excluded four patients with hemochromatosis whose high hepatic iron content impaired the MRI-PDFF measurement.

The liver status was assessed with QUS or SWE in 284 patients during the study period. After excluding 91 subjects with incomplete or invalid ultrasound scans and 52 patients who did not have a valid quantitative MRI scan, the final study cohort included 90 patients with clinically suspected NAFLD and 51 patients with suspected MAFLD.

### 2.2. Ultrasound

Patients fasted at least four hours before the ultrasound scan. All ultrasound examinations were performed with a Samsung RS85 Prestige (Samsung Medison Co., Ltd., Hongcheon, Republic of Korea) scanner and a CA 1-7S curvilinear probe. Patients were scanned in a supine position with an elevated right arm. The probe was placed into a right intercostal space perpendicular to the liver capsule, and measurements were acquired during breath holds with shallow inspirations. All scans were performed by one of three experts with at least five years of experience in liver ultrasounds. The examiners were blinded from the patients’ medical history and the MRI-PDFF results.

The QUS protocol included tissue attenuation imaging (TAI^TM^) and tissue scatter distribution imaging (TSI^TM^) for measuring AC and BSC-D, respectively. During QUS, regions of interest (ROIs) were positioned by the examiner in the right lobe of the liver at a depth of approximately 2 cm below the liver capsule, avoiding large vessels. TAI values with R^2^ < 0.6 were considered non-reliable and were discarded. TAI and TSI were calculated as the average of five valid measurements. TAI was reported in dB/cm/MHz and TSI in arbitrary units.

A 2D SWE was performed simultaneously with QUS, as it has been described previously [19]. For the LS measurement, the Shear-Wave Imaging^TM^ application was selected. A fan-shaped measurement window was positioned by the examiner in the right liver lobe at least 2 cm below the liver capsule, avoiding large vessels. The software automatically generated two color-coded maps, which displayed the distribution of LS and the reliable measure index (RMI) within the window. LS was measured within circular ROIs manually positioned in areas with the highest RMI. Measurements with an RMI <0.4 were considered non-reliable and were discarded. LS was calculated as the average of at least five valid measurements with an interquartile range/median (IQR/med.) ratio of <30%. LS was reported in kPa units. Patients were also classified into fibrosis grades using cutoff values for F2 at ≥5 kPa, for F3 at ≥9 kPa, and for F4 at ≥13 kPa based on the “rule of four” recommendation [20]. 

### 2.3. Magnetic Resonance Imaging

We used quantitative MRI as the reference standard of the fat fraction quantification. MRI was performed within one month of QUS. All patients were scanned with a 1.5 T Philips Ingenia^TM^ MRI scanner (Philips Healthcare, Amsterdam, The Netherlands) and a Q-Body coil. A 2D multi-echo gradient echo sequence was acquired at the level of the porta hepatis, with 12 evenly spaced echoes with echo time (TE) starting from 1.2 msec and increasing by 1.2 msec, repetition time (TR) of 120 msec, flip angle (FA) of 20 degrees, pixel bandwidth (Bw) of 2712 Hz, a field of view of typically 400 × 350 mm, a reconstruction matrix of 128 × 116 pixels, slice thickness and an interslice gap of 10 mm. Three circular ROIs were selected in the right lobe of the liver on the magnitude images with the MRQuantif (https://imagemed.univ-rennes1.fr/en/mrquantif (accessed on 16 September 2022)) software [21]. The software calculated the R2* and the MRI-PDFF of the liver by fitting an exponential signal decay model corrected for signal variations caused by the six main fat peaks described by Hamilton et al. [22]. Patients were classified into four severity grades using thresholds at 5%, 10%, and 20% MRI-PDFF, which have been previously used in multiple studies to diagnose mild, moderate, and severe steatosis [14,17,23].

### 2.4. Statistical Analysis

The distribution parameters were calculated for clinical and imaging variables in the dataset. Continuous variables were reported as means ± standard deviation (SD), and categorical variables as a number and percentage. We used the Kruskal–Wallis test to compare MRI-PDFF values and QUS parameters between different grades of liver fibrosis and to compare LS and BMI values between MRI-PDFF-defined steatosis grades.

We constructed univariable and multivariable linear regression models to identify ultrasound and clinical biomarkers significantly associated with the severity of fatty liver disease in the NAFLD and MAFLD groups. In the multivariable models, AC, BSC-D, LS, and BMI were included as independent variables, while MRI-PDFF was the dependent variable. In the univariable models, AC, BSC-D, LS, and BMI were separately tested against MRI-PDFF.

We created three non-linear regression models to predict MRI-PDFF based on AC, BSC-D, or the combination of both in the NAFLD dataset. The models were optimized with a least squares analysis using the following formula for each of the QUS parameters: MRI-PDFF = β1 × exp(β2 × QUS) + β3. The fitted models were validated in a *k*-fold (*k* = 3) cross-validation repeated five times on the NAFLD dataset and were also tested on the MAFLD dataset. We compared the models’ performance by calculating the coefficients of determination (R^2^), which were adjusted for the number of independent variables, and root mean square errors (RMSE) during cross-validation. We used the least squares model that combined AC and BSC-D to calculate UEFF.

Pearson’s correlation coefficients (ρ), together with the 95% confidence intervals (CI), were calculated between UEFF and MRI-PDFF in the NAFLD and MAFLD datasets. We also determined the intercept ©, regression slope (β), and significance of a simple linear model that used UEFF to predict MRI-PDFF.

We compared the predicted UEFF between the steatosis grades with the Kruskall–Wallis test. We performed a post hoc analysis with Dunn’s test, and we adjusted *p*-values with Holm’s method to prevent false discoveries from multiple comparisons. We constructed box plots to visually examine the distribution of UEFF across the four steatosis grades.

The diagnostic performance of the UEFF in between <5% vs. ≥5%, as well as between <10% vs. ≥10% MRI-PDFF, was evaluated with receiver operating characteristics (ROC) analyses. We calculated the area under the curve (AUC), together with the CI, and determined the “best” diagnostic threshold with the highest J value (J = true positive rate—false negative rate). We also calculated the sensitivity (SN), specificity (SP), negative predictive value (NPV), positive predictive value (PPV), and total accuracy (TA) for the classifications. We performed a power analysis for each binary classification problem to ensure that the sample size is large enough to keep the type 2 error <10%. 

We applied a *p* < 0.05 cutoff for all statistical tests to declare significance. The data analysis was performed in the R version 4.2.3 (www.r-project.org, accessed on 20 April 2023) statistical computing environment.

## 3. Results

### 3.1. Demographics and Characteristics of Patient Groups

We examined 141 patients with QUS and SWE to determine the severity of fatty liver and fibrosis in suspected diffuse liver disease. The etiology of the liver disease based on the clinical background was NAFLD in 90 cases and MAFLD in 51 cases. All participants had an MRI scan within one month of the US examination for measuring MRI-PDFF. All participants were of European descent.

The patients were classified into steatosis grades using cutoff values at 5%, 10%, and 20% MRI-PDFF. In the NAFLD group, 19 (21.1%) patients had <5%, 22 (24.4%) 5–10%, 34 (37.8%) 10–20%, and 15 (16.7%) ≥20% MRI-PDFF (Table 2). The distribution of patients in the MAFLD group among the same steatosis grades was 28 (54.9%), 8 (15.7%), 10 (19.6%), and 5 (9.8%), respectively. The SWE indicated significant liver fibrosis with LS ≥ 9 kPa in 14 (15.6%) NAFLD and 21 (41.2%) MAFLD patients.

The MRI-PDFF and the QUS parameters were not significantly different between the fibrosis grades in either etiology. Similarly, patients in MRI-PDFF-defined steatosis grades did not have significantly different LS.

The BMI was significantly different between <5% (mean ± SD, 26.5 ± 4.43) vs. >5–10% (29.9 ± 4.17, *p* < 0.01), 10–20% (29.4 ± 4.56, *p* < 0.008), and ≥20% (30.7 ± 3.44, *p* < 0.001) MRI-PDFF in the NAFLD group. However, BMIs did not show significant differences among the 5–10%, 10–20%, and ≥20% grades.

### 3.2. Regression Models

The regression analysis identified AC as a strong independent predictor of MRI-PDFF in both NAFLD and MAFLD groups (Table 3). BSC-D was not an independent predictor in the multivariable analysis but showed a significant association with MRI-PDFF in the univariable analysis. AC and BSC-D were better predictors of MRI-PDFF in both NAFLD and MALD than BMI, which was only weakly associated with MRI-PDFF in the univariable analysis. Meanwhile, LS did not show a significant association with liver fat. 

We constructed scatter plots to visualize the correlation between QUS biomarkers and MRI-PDFF (Figure 1). The relationship between AC, BSC-D, and MRI-PDFF was non-linear, with measurements gradually plateauing above a 20% fat fraction. This effect was more pronounced in the case of BSC-D than AC. The distribution of the QUS parameters with respect to the fat fraction was similar in the NAFLD and MAFLD groups.

### 3.3. Non-Linear Least Squares Models

We developed three models using a non-linear least squares analysis for the prediction of MRI-PDFF based on QUS. The model parameters were fitted on either AC or BSC-D or both in the NAFLD group, and the accuracy of the three models was validated in a three-fold cross-validation repeated five times. The AC model was the best fit for the validation set (mean ± SD, adj. R^2^ = 0.498 ± 0.046) with a mean RMSE of 6.415% ± 0.561% in the cross-validation (Table 4). The combined AC and BSC-D model yielded only a slightly lower coefficient of determination (adj. R^2^ = 0.446 ± 0.152) and RMSE (RMSE = 7.347% ± 3.105%). The BSC-D model performed the worst with a mean adj. R^2^ of 0.318 ± 0.075 and RMSE of 7.687% ± 0.696%

We also evaluated the same three models in the MAFLD dataset (Table 4). The AC model (adj. R^2^ = 0.733) was the best fit, followed by the combined (adj. R^2^ = 0.673) and the BSC-D (adj. R^2^ = 0.253) models. The RMSE of the AC, combined, and BSC-D models were 3.911%, 4.574%, and 6.608%, respectively.

### 3.4. Correlation between UEFF and MRI-PDFF

We chose the combined model to predict UEFF in both the NAFLD and MAFLD datasets. The mean and SD of the UEFF (12.06% ± 7.23%) were similar to the metrics calculated for MRI-PDFF. The correlation between UEFF and MRI-PDFF for all of cases in the NAFLD group was strong (ρ = 0.735, CI = 0.622–0.817, *p* < 0.001) (Figure 2). However, the correlation was the strongest (ρ = 0.886, CI = 0.81–0.933, *p* < 0.001) in the range of 0–12.2% MRI-PDFF. The slope of the regression line between UEFF and MRI-PDFF was β = 0.9 with a significance of *p* < 0.001, and the intercept was at c = 1.238%.

In the MAFLD dataset, the mean and SD of UEFF (7.08% ± 7.86%) were nearly identical to those calculated for MRI-PDFF. The UEFF was in a very strong correlation (ρ = 0.828, CI = 0.716–0.899, *p* < 0.001) with MRI-PDFF, peaking (ρ = 0.902, CI = 0.813–0.95, *p* < 0.001) between 0.7% and 14.5%. The regression slope was β = 0.81 (*p* < 0.001) with a regression intercept of c = 2.06%.

Meanwhile, when only patients with severe steatosis (≥20% MRI-PDFF) were examined, the correlation between the UEFF and MRI-PDFF was not significant in either the NAFLD (ρ = 0.227, CI =−0.323–0.662, *p* < 0.416) or the MAFLD (ρ = 0.091, CI = −0.9–0.86, *p* < 0.884) group.

### 3.5. Comparison of UEFF between Steatosis Grades

We divided patients into four grades (none (<5%), mild (5–10%), moderate (10–20%), and severe (≥20%)) based on the severity of steatosis measured with MRI-PDFF. We calculated the mean and SD of the UEFF predicted using the combined least squares model for each grade. In the NAFLD group, the UEFF was significantly different between <5% (2.70% ± 2.48%, *p* < 0.0048) and 5–10% (10.1% ± 4.03%), as well as between 5–10% and 10–20% (15.3% ± 4.45%, *p* < 0.006) MRI-PDFF steatosis (Figure 3). In the MAFLD group, the UEFF was significantly different between none (1.65% ± 2.04%, *p* < 0.001) and mild (9.12% ± 3.6%) steatosis; meanwhile, the difference was non-significant between mild and moderate (14.1% ± 5.3%) and between moderate and severe (20.3% ± 10.1%) steatosis.

### 3.6. Diagnosis of Different Grades of Steatosis with UEFF

The ROC analysis showed that UEFF could separate ≥5% from <5% MRI-PDFF, with excellent accuracy (AUC = 0.971, CI = 0.942–1) in the NAFLD group. The best cutoff was at 6.36%, achieving a sensitivity of 0.92, specificity of 1, NPV of 0.76, PPV of 1, and TA of 0.93 (Table 5). The UEFF was similarly successful in differentiating between mild (≥5% MRI-PDFF) vs. no steatosis (AUC = 0.99, CI = 0.973–1) in MAFLD (Figure 4). The sensitivity, specificity, NPV, PPV, and TA of the classification were all 0.96, when using a diagnostic threshold of 5.22% (Figure 4).

For detecting ≥10% MRI-PDF in NAFLD, the AUC for UEFF was 0.91 (CI = 0.848–0.972), and sorting with an 11.64% threshold yielded a sensitivity, specificity, NPV, PPV, and TA of 0.86, 0.88, 0.89, 0.84, and 0.87, respectively. Meanwhile, in the MAFLD group, the AUC for diagnosing ≥10% MRI-PDFF was 0.961 (CI = 0.913–1). The optimal UEFF cutoff was at 8.74, which could predict moderate/severe steatosis with a sensitivity, specificity, NPV, PPV, and TA of 0.93, 0.94, 0.88, 0.97, and 0.94, respectively.

## 4. Discussion

We fitted a model using non-linear least squares analysis to predict UEFF based on AC and BSC-D parameters as the input, and MRI-PDFF as the reference standard in 90 NAFLD cases. We found that the correlation between the predicted UEFF and MRI-PDFF was strong (ρ = 0.735), and that the UEFF model was able to diagnose ≥5% (AUC = 0.97) and ≥10% MRI-PDFF (AUC = 0.91) steatosis with an excellent accuracy in NAFLD. We have also validated our UEFF model in multiple ways. We performed a three-fold cross-validation that was repeated five times on the NAFLD dataset, which showed that the AC, combined AC, and BSC-D models were the best fit on the NAFLD dataset, producing nearly identical mean coefficients of determination (adj. R^2^ = 0.498 vs. 0.446). We also tested our models on a dataset collected from patients with MAFLD with various etiologies of CLD. Our results show that the UEFF model developed for patients with NAFLD can also be applied to predict steatosis in MAFLD with good accuracy. The coefficient of determination for the UEFF model trained on the NAFLD dataset did not decrease in the MAFLD dataset (adj. R^2^ = 0.673). The predicted UEFF very strongly correlated with MRI-PDFF (ρ = 0.828) and could detect patients with ≥5% (AUC = 0.99) and ≥10% (AUC = 0.96) MRI-PDFF with nearly perfect accuracy. Thus, our results suggest that a UEFF model can be implemented for diagnosing low- and moderate-grade hepatic steatosis with similar accuracy to MRI-PDFF in both NAFLD and MAFLD.

A handful of studies have already reported prediction models for estimating the hepatic fat fraction based on a selection of QUS parameters [14,17]. Another approach to developing an ultrasonic fat fraction estimator is to use convolutional neural networks (CNNs) trained on either radiofrequency or image data [24,25]. The correlation between the predicted fat fraction and MRI-PDFF (ρ = 0.76–0.87) and the classification accuracy for low-grade (≥5% MRI-PDF) steatosis (AUC = 0.89–0.98) reported for these models was similar to the results of our model. Notably, thresholds for diagnosing ≥5% (6.36% vs. 6.34%) and ≥10% (11.64% vs. 11.7%) MRI-PDFF with the UEFF model in NAFLD were almost identical to the cutoff values reported for the UDFF model [17]. The novelty of our approach was that we applied a non-linear least squares analysis in AC measured with the TAI and BSC-D measured with the TSI applications. This is important since BSC-D measured with a different technique had a higher coefficient of determination in a least squares analysis compared to TSI (R^2^ = 0.76 vs. R^2^ = 0.32) [17]. In addition, all prior studies were performed in a single-center setting and only analyzed the QUS data from patients with NAFLD without an external validation set, which may raise concerns about overfitting the models. We tested the UEFF model on the data collected from the patients with MAFLD and did not see a drop in the model’s performance, suggesting that our model does not overfit on the NAFLD dataset.

Previous studies have already demonstrated the saturation of QUS parameters in high-grade steatosis [17,24,25]. The correlation between MRI-PDFF and the predicted fat fraction was linear and below 18% MRI-PDFF; outside this range, the correlation decreased, and the prediction models underestimated the severity of steatosis [24,25]. We found that the correlation between UEFF and MRI-PDFF was the highest (ρ = 0.89) in the range of <12.2% MRI-PDFF in the NAFLD group. Meanwhile, the overall correlation between UEFF and MRI-PDFF was also strong in both the patients with NAFLD (ρ = 0.735) and MAFLD (ρ = 0.828), which is an advantage of our method compared to CAP, showing a lower overall correlation (ρ = 0.44–0.577) with MRI-PDFF in previous studies [23,26]. The higher correlation and better prediction accuracy observed in the MAFLD group can also be explained, in part, by the saturation effects as the percentages of patients with ≥10% (16.7% vs. 37.8%) and ≥20% (9.8% vs. 19.6%) MRI-PDFF were lower compared to the NAFLD group. When the correlation was calculated selectively for the patients with severe (≥20% MRI-PDFF) steatosis, it was not significant in either the NAFLD or the MAFLD group. A similar loss of correlation was reported for CAP at ≥331 dB/m, corresponding to ≥21.4% MRI-PDFF [26]. In our opinion, this finding does not significantly impact the clinical application of our method since the primary task of UEFF is to diagnose liver steatosis in an early stage and identify patients with moderate- or higher-grade steatosis (≥10% MRI-PFF) who have an increased risk of mortality rather than providing exact measurements of the liver fat fraction at ≥20%.

We also examined whether liver fibrosis had a confounding effect on QUS or MRI-PDFF. We did not see that LS had a significant influence on either MRI-PDFF or other QUS parameters. LS was not significantly different between MRI-defined steatosis grades; also, MRI-PDFF, AC, and BSC-D did not differ between the fibrosis grades. Furthermore, LS was not a significant predictor of MRI-PDFF in a linear regression. AC has been extensively investigated as an imaging biomarker of hepatic steatosis in a multitude of publications [12,27]. A recent prospective study has reported that, in a cohort of 124 patients, a QUS-derived fat fraction was more accurate for diagnosing NAFLD-related hepatic steatosis than the controlled attenuation parameter (CAP) [23]. Another study, which used a 2D CNN to predict the fat fraction in 173 NAFLD patients, also found that the accuracy of the CNN model was superior to AC for the detection of steatosis [25]. Interestingly, among the least squares models built from our QUS data, the simple AC model had a slightly better mean coefficient of determination (adj. R^2^ = 0.498 vs. 0.446) and lower RMSE (6.42% vs. 7.35%) in the cross-validation than the combined model. The goodness-of-fit was also better with the AC model in the MAFLD group (adj. R^2^ = 0.73 vs. 0.67). These results may imply that a solely AC-based model does not significantly underperform the combined model while being less susceptible to overfitting due to the higher number of trainable parameters; thus, it could be more generalized. Meanwhile, the addition of a second variable, such as BSC-D, to the model may improve the robustness of the prediction by compensating for the noise in the measurements. These questions should be further addressed in large-scale multi-center studies.

Our study had several limitations. First, the data collection was retrospective, and relatively small patient cohorts with clinically suspected NAFLD (90 cases) and MAFLD (51 cases) were examined in a single center. Therefore, our results cannot be generalized to large patient populations. Second, there was a selection bias towards steatosis, as high-risk patients were preselected during clinical screening. Therefore, the percentage of negative cases in the NAFLD (21.2%) and MAFLD (54.9%) groups does not represent the general population. Third, AC and BSC-D were measured with the TAI and TSI applications, which might have slightly different technical specifications compared to similar QUS parameters available from different vendors.

## 5. Conclusions

With the recent advancements in image analysis, new algorithmic- and artificial intelligence-based methods have become available for the objective assessment of both diffuse and focal liver disease [28]. The prevalence of FLD is steadily increasing worldwide, which necessitates the introduction of new quantitative imaging techniques that can be used for an objective and early diagnosis of hepatic steatosis [1]. In this study, we have developed a new, easily interpretable biomarker, UEFF, for the quantification of steatosis based on QUS parameters. We have demonstrated that UEFF is capable of diagnosing NAFLD with an accuracy similar to MRI-PDFF. Moreover, the UEFF also showed a high degree of agreement with MRI-PDFF in patients with MAFLD. Thus, UEFF may be an ideal screening tool for FLD; however, it requires further validation in large-scale, multi-center studies.

## Figures and Tables

**Figure 1 diagnostics-13-03353-f001:**
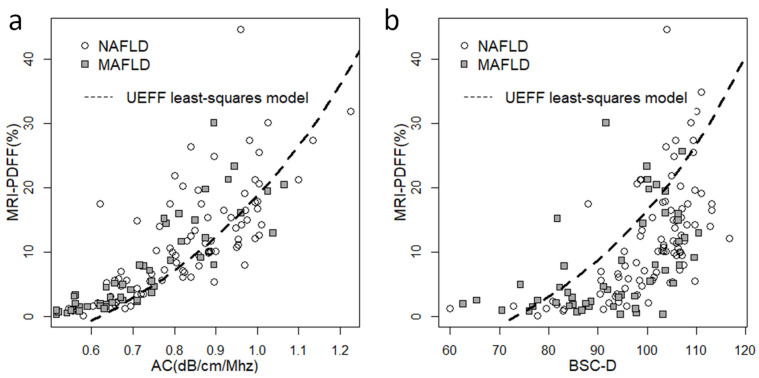
Scatter plots showing the correlation between quantitative ultrasound (QUS) biomarkers and MRI-PDFF. (**a**) We observed a non-linear distribution of the attenuation coefficient (AC) and (**b**) the backscatter-distribution coefficient (BSC-D) in relation to MRI-PDFF, in both the non-alcoholic fatty liver disease (NAFLD, empty circle) and the metabolic-associated fatty liver disease (MAFLD, gray square) groups. We fitted a non-linear least squares model (dashed line) onto the NAFLD dataset using a combination of AC and BSC-D values to compensate for the saturation effect at higher fat fractions.

**Figure 2 diagnostics-13-03353-f002:**
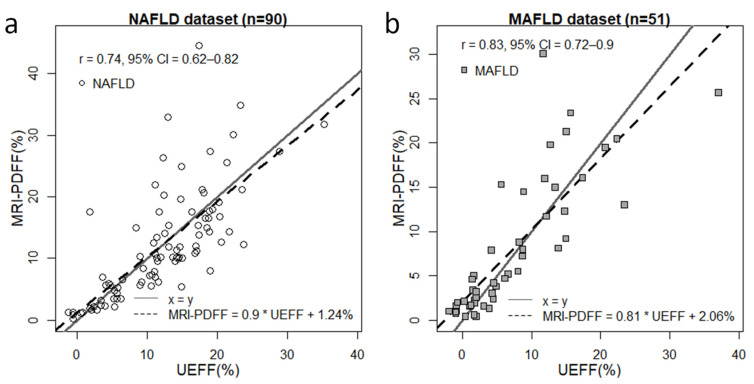
Correlation between UEFF and MRI-PDFF. The ultrasound-estimated fat fraction (UEFF) was predicted with a combined least squares model fitted on the attenuation coefficient (AC) and backscatter-distribution coefficient (BSC-D). (**a**) The correlation between UEFF and MRI-PDFF was strong (Pearson’s r = 0.73, 95% confidence interval (CI) = 6.22–0.817) in the non-alcoholic fatty liver disease (NAFLD) group. (**b**) We found a very strong correlation between UEFF and MRI-PDFF (r = 0.828, CI = 0.716–0.899) by applying the same model to the metabolic-associated fatty liver disease (MAFLD) group. The slope and the offset of the correlation line (dashed line) were 0.9 and 1.24% in the NAFLD and 0.81 and 2.06% in the MAFLD dataset. The x = y identity line (continuous gray line) is shown as a reference.

**Figure 3 diagnostics-13-03353-f003:**
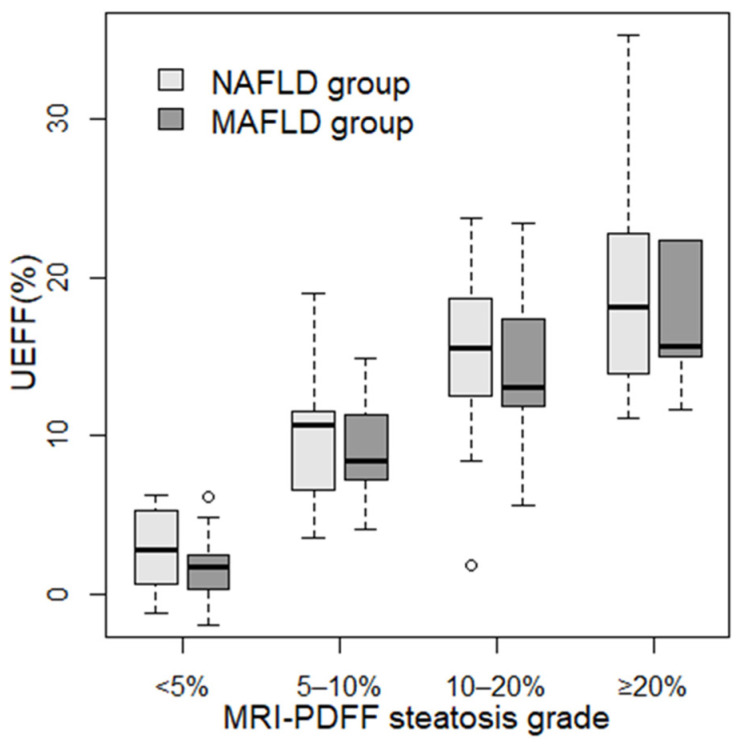
Distribution of UEFF across steatosis grades. Patients were classified into four steatosis severity grades based on MRI-PDFF using cutoff values at 5%, 10%, and 20% fat fractions. The box plots represent the median (thick line), the range between 25% and 75% percentiles (box), and the minimum and maximum values (whiskers) of UEFF predicted with the combined least squares model for patients in each grade. Two sets of box plots were calculated, one for non-alcoholic fatty liver disease (NAFLD, light gray), and another for metabolic-associated fatty liver disease (MAFLD, dark gray).

**Figure 4 diagnostics-13-03353-f004:**
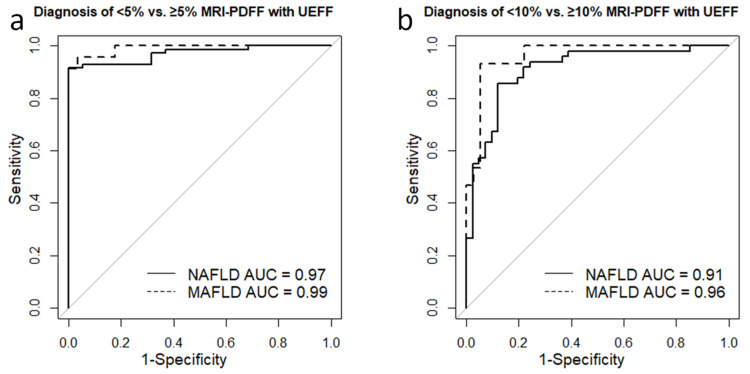
Diagnostic performance of UEFF in separating steatosis grades. (**a**) The receiver operating characteristics (ROC) analysis showed that UEFF could separate <5% from ≥5% MRI-PDFF steatosis with excellent area under the ROC curve (AUC) accuracy in both non-alcoholic fatty liver disease (NAFLD, AUC = 0.97, continuous line) and metabolic-associated fatty liver disease (MAFLD, AUC = 0.99, dashed line). (**b**) The UEFF proved to be just slightly less successful in differentiating between <10% vs. ≥10% MRI-PDFF with an AUC of 0.91 in NAFLD (continuous line) and an AUC of 0.96 in MAFLD (dashed line).

**Table 1 diagnostics-13-03353-t001:** Demographics and etiology of the patient groups.

	NAFLD (*n* = 90)	MAFLD (*n* = 51)
**Sex**		
female (%)	44 (48.9%)	26 (51.0%)
male (%)	46 (51.1%)	25 (49.0%)
**Age (years)**		
mean (±SD)	55.1 (13.3)	54.3 (13.4)
range	23–78	21–82
**BMI (kg/m^2^)**		
mean (±SD)	29.3 (4.4)	26.6 (3.9)
range	18.6–45.3	19.7–35.3
**T2DM (%)**	24 (35.3%)	1 (2.7%)
**Etiology of chronic liver disease (%)**		
AIH		5 (9.8%)
alcoholic		3 (5.9%)
chr. HBV		3 (5.9%)
chr. HCV		7 (13.7%)
hemochr.		6 (11.8%)
hepatotoxic med.		15 (29.4%)
PBC/PSC		8 (15.7%)
Wilson’s disease		4 (7.8%)
none	91 (100%)	

AIH: autoimmune hepatitis, BMI: body mass index, chr.HBV: chronic hepatitis B, chr.HCV: chronic hepatitis C, hemochr.: hemochromatosis, med.: medication, PBC: primer biliary cholangitis, PSC: primer sclerosing cholangitis, T2DM: type 2 diabetes mellitus.

**Table 2 diagnostics-13-03353-t002:** Distributions of the imaging parameters.

	NAFLD (*n* = 90)	MAFLD (*n* = 51)
**AC (dB/cm/Mhz)**		
mean ± SD	0.83 ± 0.15	0.73 ± 0.17
range	0.55–1.22	0.49–1.26
**BSC-D (arbitrary unit)**		
mean ± SD	100.54 ± 9.6	92.68 ± 11.87
range	60–116.62	62.52–110.38
**LS (kPa)**		
mean ± SD	7.24 ± 3.2	11.26 ± 8.9
range	3.2–21.8	3.9–39.7
**MRI-PDFF (%)**		
mean ± SD	12.1 ± 8.86	7.8 ± 7.7
range	0.2–44.6	0.4–30.1
**UEFF (%) ^1^**		
mean ± SD	12.06 ± 7.23	7.08 ± 7.86
range	−1.25–35.3	−1.99–37.03
**Steatosis grade ^2^**		
<5%	19 (21.1%)	28 (54.9%)
5–10%	22 (24.4%)	8 (15.7%)
10–20%	34 (37.8%)	10 (19.6%)
≥20%	15 (16.7%)	5 (9.8%)
**Fibrosis grade ^3^**		
F0/F1	12 (13.3%)	5 (9.8%)
F2	64 (71.1%)	25 (49.1%)
F3	8 (8.9%)	12 (23.5%)
F4	6 (6.7%)	9 (17.6%)

^1^ The UEFF was calculated with the least squares model that combined AC and BSC-D to predict MRI-PDFF. ^2^ The steatosis grade was calculated from MRI-PDFF using cutoff values at 5%, 10%, and 20%. ^3^ The fibrosis grade was determined from LS using thresholds according to the “rule of four” at 5 kPa, 9 kPa, and 13 kPa [20]. AC: attenuation coefficient, BSC-D: backscatter-distribution coefficient, LS: liver stiffness, MAFLD: metabolic-associated fatty liver disease group, MRI-PDFF: magnetic resonance imaging-detected fat fraction, NAFLD: non-alcoholic fatty liver disease group, UEFF: ultrasound-derived fat fraction.

**Table 3 diagnostics-13-03353-t003:** Results of the regression analysis.

	Univariable Analysis				Multivariable Analysis			
	adj. R^2^	F Stat.	ß	*p*-Value	adj. R^2^	F Stat.	ß	*p*-Value
NAFLD								
AC	0.504	91.47	43.04	<0.001	0.485	18.65	36.16	<0.001
BSC-D	0.31	14.01	0.52	<0.001	0.485	18.65	0.18	0.118
LS	0.0009	1.08	0.3	0.3	0.485	18.65	0.06	0.781
BMI	0.078	7.36	0.63	0.008	0.485	18.65	0.07	0.696
MAFLD								
AC	0.748	149.2	40.19	<0.001	0.775	34.49	44.82	<0.001
BSC-D	0.235	16.38	0.32	<0.001	0.775	34.49	0.07	0.298
LS	0.049	3.57	−0.22	0.065	0.775	34.49	0.05	0.521
BMI	0.23	12.65	1.07	0.001	0.775	34.49	−0.21	0.38

AC: attenuation coefficient, adj.: adjusted, BMI: body mass index, BSC-D: backscatter-distribution coefficient, F stat.: F statistics, LS: liver stiffness, MAFLD: metabolic-associated fatty liver disease group, NAFLD: non-alcoholic fatty liver disease group.

**Table 4 diagnostics-13-03353-t004:** Performance of the non-linear least squares-fitted prediction models.

	NAFLD Group ^1^				MAFLD Group ^2^			
Model	adj. R^2^	RMSE	Cor. Coef.	*p*-Value	adj. R^2^	RMSE	Cor. Coef.	*p*-Value
AC	0.498 ± 0.046	6.415 ± 0.561	0.716 0.6–0.8	<0.001	0.733	3.911	0.859 0.76–0.92	<0.001
BSC-D	0.318 ± 0.075	7.687 ± 0.696	0.581 0.43–0.7	<0.001	0.253	6.608	0.518 0.28–0.69	<0.001
AC + BSC-D	0.446 ± 0.152	7.347 ± 3.105	0.735 0.62–0.82	<0.001	0.673	4.574	0.828 0.72–0.9	<0.001

^1^ The adj. R^2^ and RMSE are calculated as the mean ± standard deviation in the validation set in a three-fold cross-validation repeated five times. ^2^ The model parameters were fitted on the NAFLD group, and trained models were tested on the MAFLD group. AC: attenuation coefficient, adj. R^2^: adjusted coefficient of determination, BSC-D: backscatter-distribution coefficient, Cor. coef.: the Pearson’s correlation coefficient between the predicted fat fraction and MRI-PDFF, and the 95% confidence intervals, MAFLD: metabolic-associated fatty liver disease, NAFLD: non-alcoholic fatty liver disease, RMSE: root mean square error.

**Table 5 diagnostics-13-03353-t005:** Diagnostic performance of UEFF in separating MRI-PDFF-defined steatosis grades.

	<5% vs. ≥5% MRI-PDFF		10% vs. ≥10% MRI-PDFF	
	NAFLD Group	MAFLD Group	NAFLD Group	MAFLD Group
Cutoff	6.36%	5.22%	11.64%	8.74%
AUC	0.97	0.99	0.91	0.96
SN	0.92	0.96	0.86	0.93
SP	1	0.96	0.88	0.94
PPV	1	0.96	0.89	0.88
NPV	0.76	0.96	0.84	0.97
TA	0.93	0.96	0.87	0.94
Power	1	1	1	1

AUC: area under the curve from receiver operating curve analysis, MAFLD: metabolic-associated fatty liver disease, NAFLD: non-alcoholic fatty liver disease, NPV: negative predictive value, PPV: positive predictive value, SN: sensitivity, SP: specificity, TA: total accuracy.

## Data Availability

The data presented in this study are available on request from the corresponding author.

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
