# Peer review of "The Calculation and Evaluation of an Ultrasound-Estimated Fat Fraction in Non-Alcoholic Fatty Liver Disease and Metabolic-Associated Fatty Liver Disease"

_diagnostics, 2023, doi:10.3390/diagnostics13213353_

Round 1

Reviewer 1 Report

Comments and Suggestions for Authors

I think the non-linear regression model proposed here is interesting and useful and I recommend the publication of the paper in the present form.

Author Response

We thank Reviewer #1 for the favorable comments on our manuscript.

Reviewer 2 Report

Comments and Suggestions for Authors

Concerns

1. It has been reported that  ultrasound-estimated fat fraction ,namely CAP derived from VCTE's diagnostic ability (0.63-0.95) was lower than that of MRI-PDFF and decreased with increasing BMI compared to MRI-PDFF. MRI-PDFF is more accurate than CAP in detecting steatosis in overweight and obese patients with NAFLD.(PMCID: PMC9763419). Therefore, the authors may consider using BMI to adjust the non-linear associations between ultrasound-estimated fat fraction and MRI-PDFF.

2. From a prospective clinical trial (PMCID: PMC9315137) conducted with a Chinese cohort, this study demonstrated a correlation between CAP and MRI‐PDFF measurements for grading hepatitis steatosis; however, this correlation disappeared when CAP values were greater than 331 dB/m. So whether  MRI-PDFF could be predicted by CAP  when CAP  was greater than 331 dB/m deserves further discussion,

3. The single center design and limmitted sample size would greatly attenuate its evidence of supporting cocnlusion.

Comments on the Quality of English Language

 Minor editing of English language required.

Author Response

  1. It has been reported that  ultrasound-estimated fat fraction ,namely CAP derived from VCTE's diagnostic ability (0.63-0.95) was lower than that of MRI-PDFF and decreased with increasing BMI compared to MRI-PDFF. MRI-PDFF is more accurate than CAP in detecting steatosis in overweight and obese patients with NAFLD.(PMCID: PMC9763419). Therefore, the authors may consider using BMI to adjust the non-linear associations between ultrasound-estimated fat fraction and MRI-PDFF.

We agree with the reviewer that MRI-PDFF is the most accurate method to assess liver steatosis in extremely obese patients and severe steatosis characterized by marked attenuation and scatter of the ultrasound signal. Our data clearly demonstrate the saturation of the QUS parameters at higher grades of steatosis. On the other hand, we do not think that including BMI in our UEFF model would improve its accuracy for multiple reasons. First, we trained our UEFF model on the NAFLD dataset, and based on linear regression analysis BMI only very weakly correlated with and was not an independent predictor of MRI-PDFF in this group (please refer to Table 3). Second, we compared BMI between MRI-defined steatosis grades, and although BMI was significantly different between subjects with <5% vs. ≥ 5% MRI-PDFF, the comparison was not significant between higher grades based on the Kruskal-Wallis and post-hoc Dunn tests (please see Results, page 12, lines 240-243). 

  1. From a prospective clinical trial (PMCID: PMC9315137) conducted with a Chinese cohort, this study demonstrated a correlation between CAP and MRI‐PDFF measurements for grading hepatitis steatosis; however, this correlation disappeared when CAP values were greater than 331 dB/m. So whether  MRI-PDFF could be predicted by CAP  when CAP  was greater than 331 dB/m deserves further discussion.

We agree with the reviewer that the correlation between UEFF and MRI-PDFF is stronger at lower grades of steatosis. Actually, we have detected the strongest correlation between the two methods in the range of 0 – 12.2% MRI-PDFF. However, the overall correlation between UEFF and MRI-PDFF was also strong in both NAFLD (Pearson`s r =0.735) and MAFLD (r=0.828) patients (please see Results, page 18, lines 308-316). We think that this is an advantage of our method compared to CAP, which showed a lower overall correlation (r = 0.577) with MRI-PDFF in the study recommended by the reviewer.  We have also calculated the correlation between USFF and MRI-PDFF selectively for cases with high grade (≥20% MRI-PDFF) of steatosis, as the reviewer suggested. Similar to CAP, the correlation was not significant in either the NAFLD (r = 0.227, p<0.416) or the MAFLD (r = 0.091, p<0.884) group (please see Results, page 18, lines 318-320). We think this finding does not significantly impact the clinical application of our method since the primary goal for UEFF is to diagnose liver steatosis in an early stage and identify patients with moderate or higher grade steatosis (≥10% MRI-PFF) who have an increased risk of mortality rather than providing exact measurements of liver fat fraction at ≥20% (please refer to Discussion, page 25, lines 429-436).

  1. The single center design and limmitted sample size would greatly attenuate its evidence of supporting cocnlusion.

  We agree with the reviewer that large-scale,  multi-center studies are needed for the comprehensive evaluation of QUS-based fat quantification. The limitations of our study due to its single-center design and the limited number of subjects have already been discussed in the manuscript (please see Discussion, page 26, lines 457-464).

Reviewer 3 Report

Comments and Suggestions for Authors

Dear Editor,

Having reviewed the submitted manuscript detailing the use of a non-linear least squares analysis model for predicting the ultrasound fat fraction (USFF) based on MRI-PDFF in NAFLD cases, I would like to commend the authors on their well-conducted research and insightful findings. The methodologies adopted, the data presented, and the subsequent analyses are both rigorous and comprehensive.

The focus on comparing the efficiency of this predictive model against the widely-recognized MRI-PDFF and its potential applicability in both NAFLD and MAFLD makes this a significant contribution to the field. The discussion, in particular, offers a deep dive into the intricate details, comparing this study's outcomes with previously established models, which enriches the literature.

Given the merits of the work, I would highly recommend this manuscript for publication.

However, for the benefit of the academic community and to further enhance the impact of this research, I'd like to suggest the following directions for future research:

1.     Multi-Center Trials: Considering the model's promise, it would be pivotal to test its applicability and reliability in a multi-center setting. This would provide insights into potential device-related or patient demographic biases and solidify the model's potential for widespread use.

2.     Diverse Patient Demographics: To truly gauge the generalizability of this model, a more extensive and diverse patient demographic should be studied. Such an analysis would ensure that the model can be applied universally, catering to various patient backgrounds.

3.     Direct Model Comparisons: The manuscript already offers comparative insights into the efficiency of this model against others. It would be intriguing to directly pit the model against newer methodologies, such as those based on convolutional neural networks (CNNs), in a head-to-head comparison to discern their relative efficacies.

4.     Long-Term Efficacy: Engaging in longitudinal studies would be beneficial to understand the model's long-term accuracy, potential drifts in predictions, and its reliability over extended periods.

5.     Technological Disparities: Given the potential for disparities arising from varying technical specifications across different vendors, a dedicated study or subsection analyzing these differences could be highly enlightening.

6.     Incorporating Advanced Algorithms: With rapid advancements in AI, integrating the latest algorithms into this predictive model or comparing its efficiency against these newer algorithms would be a promising avenue of research.

Once again, I'd like to congratulate the authors on their commendable work and look forward to seeing this research published and further contributions in the future.

Author Response

We thank Reviewer#3 for the favorable comments on our manuscript and for the excellent suggestions regarding future research directions.

Reviewer 4 Report

Comments and Suggestions for Authors

I have no comments. Well-written and interesting article.

Author Response

We thank Reviewer#4 for finding our paper valuable and interesting.